# The Construction of a High-Density Genetic Map for the Interspecific Cross of *Castanea mollissima* × *C. henryi* and the Identification of QTLs for Leaf Traits

Xibing Jiang [1,2] , Yanpeng Wang [1,*] , Junsheng Lai [3], Jian Wu [3], Conglian Wu [3], Weiyun Hu [3], Xiaolong Wu [3] and Bangchu Gong [1,*]

1   Research Institute of Subtropical Forestry, Chinese Academy of Forestry, Hangzhou 311400, China; jxb912@caf.ac.cn
2   State Key Laboratory of Tree Genetics and Breeding, Chinese Academy of Forestry, Beijing 100091, China
3   Qingyuan Bureau of Natural Resources and Planning, Lishui 323800, China; 17858909639@163.com (W.H.)
*   Correspondence: yanpengwang@caf.ac.cn (Y.W.); gongbc@126.com (B.G.); Tel.: +86-571-63131182 (B.G.); Fax: +86-571-63341304 (B.G.)

**Abstract:** Chinese chestnut is an economically and ecologically valuable tree species that is extensively cultivated in China. Leaf traits play a vital role in the photosynthetic capacity, chestnut yield, and quality, making them important breeding objectives. However, there has been limited research on constructing high-density linkage maps of Chinese chestnut and conducting quantitative trait loci (QTL) analyses for these leaf traits. This knowledge gap has hindered the progress of selection in Chinese chestnut breeding. In this study, we selected a well-established interspecific $F_1$ population, consisting of *Castanea mollissima* 'Kuili' × *C. henryi* 'YLZ1', to construct comprehensive genetic maps for chestnut. Through the use of a genotyping-by-sequencing (GBS) technique, we successfully created a high-density linkage map based on single-nucleotide polymorphisms (SNPs) from the $F_1$ cross. The results showed that 4578 SNP markers were identified in the genetic linkage map, and the total length was 1812.46 cM, which was distributed throughout 12 linkage groups (LGs) with an average marker distance of 0.4 cM. Furthermore, we identified a total of 71 QTLs associated with nine chestnut leaf traits: chlorophyll b content (chlb), stomatal conductance (Gs), leaf area (LA), leaf dry weight (LDW), leaf fresh weight (LFW), leaf length (LL), leaf width (LW), petiole length (PL), and specific leaf weight (SLW). These QTLs were identified based on phenotypic data collected from 2017 to 2018. Notably, among the 71 QTLs, 29 major QTLs were found to control leaf area (LA), leaf dry weight (LDW), and leaf width (LW). The high-density genetic mapping and QTL identification related to leaf traits in this study will greatly facilitate marker-assisted selection (MAS) in chestnut breeding programs.

**Keywords:** *Castanea* spp.; interspecific hybridization; genetic map; marker-assisted breeding; QTL; leaf traits

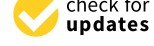


## 1. Introduction

Chestnut (*Castanea* spp., Fagaceae family) is both an economically and ecologically valuable tree in the wood processing industry that is widely distributed throughout China [1,2]. Multiple functions of chestnut have been explored in recent years; abundant secondary metabolisms, such as tannins, polyphenols, and polysaccharides extracted from the plant's inner/outer shells, spring buds, and male flowers are used in food industry, medicine, and pharmaceutical fields [3–5]. Further new roles for chestnut are being found in ecosystems and agroforestry systems [1,6,7].

Some varieties of Chinese chestnut are considered to show perfect performance in resisting different diseases and pests, such as *Phytophthora cinnamomi* (*P. cinnamomi*), *Cryphonectria parasitica* (*C. parasitica*) and the chestnut gall wasp [8–10]. Chestnut germplasm

resources are abundant, and have been useful tools in breeding new varieties to improve their ability to resist biotic or abiotic stresses in European and American countries [11]. However, progress in chestnut breeding is severely hindered because of the plant's longer juvenile period and conventional breeding when compared with annual plants. Identification of quantitative trait loci (QTL) mainly relies on the development of linkage and association maps, key genes associated with plant traits, and the development of MAS, which could prevent these defects [12–14].

The development of genetic markers and their applications is highly appreciated due to their efficiency and ability to solve problems [12,15–18]. Moreover, through the utilization of NGS technologies, a high number of SNP markers could be generated using genotyping-by-sequencing (GBS) approaches, which could allow for the construction of high-density genetic maps [12,19,20]. However, more anticipated markers and higher-density genetic maps are still needed for QTL mapping, and their application will speed up the process of breeding chestnuts.

Casasoli et al. (2001) constructed a genetic linkage map based on RAPD, ISSR and isozyme markers, containing 381 molecular markers (311 RAPDs, 65 ISSRs, 5 isozymes) of European chestnut (*Castanea sativa* Mill.) [21]. The QTL analysis of the $F_1$ progeny of a *Castanea sativa* Mill with three different adaptive traits, and 35, 28, and 17 individual QTLs were detected for phenology, growth, and carbon isotope discrimination [22]. By using the GBS of an $F_1$ cross between 'Yanshanzaofeng' and 'Guanting No. 10' to conduct a SNP-based high-density map, Ji et al. (2018) identified 17 QTLs for five nut traits. Even though some valuable QTLs were identified based on this conducted Castanea genus gebetic map, higher-density maps and markers are still needed [19,23,24]. However, there have been few reports on the mapping of quantitative trait loci (QTLs) related to chestnut leaf traits. High-resistance chestnut varieties require the establishment of high-density genetic maps and the identification of QTLs associated with exceptional leaf traits.

The leaves of plants are the main photosynthetic organ, and are of great importance for the yield of crops. Some agronomic, phyto-physicochemical and stress-resilient traits of leaves are important in crop improvement and breeding [25]. The main objectives of breeding are to produce Chinese chestnuts with the leaf traits of a phenotype that is highly resistant to different stresses and displays good performance growth parameters that are conducive to high yield. However, QTL mapping in relation to chestnut leaf traits has not been reported yet. In order to obtain excellent leaf traits through MAS breeding, we need high-density genetic maps of Chinese chestnut and a large number of desired QTLs.

Here, we selected a established $F_1$ population derived from two *Castanea* Mill. cultivars, i.e., *C. mollissima* 'Kuili' (as the female, with a higher leaf area, total chlorophyll content, and photosynthesis capacity) and *C. henryi* 'YLZ1' (as the male, with a lower leaf area, total chlorophyll content, and photosynthesis capacity) for the construction of genetic linkage map based on identified SNP markers using GBS. Furthermore, QTL analyses were conducted using a high-density genetic map with the leaf traits data of the cross parents and the $F_1$ generation, which include the total chlorophyll content (chl a + b), chlorophyll a content (chla), chlorophyll b content (chlb), the ratio of chlorophyll a to chlorophyll b(chla/b), leaf area (LA), leaf dry weight (LDW), leaf fresh weight (LFW), leaf water content (LMC), leaf length (LL), leaf width (LW), petiole length (PL), leaf shape index (LSI), specific leaf weight (SLW), stomatic conductance(Gs), intercellular $CO_2$ concentration(Ci), net photosynthetic rate (Pn), transpiration rate (Tr), and water use efficiency (WUE). Through a traits and linkage analysis of the relationship between leaf traits, resistance and yield traits, useful insights into the next steps in the molecular breeding of Chinese chestnut can also be provided.

## 2. Material and Methods

### 2.1. Plant Materials and DNA Extraction

We selected two widely grown Chinese chestnut cultivars "Kuili" and "YLZ1", as the female and male, respectively. "Kuili" is an improved *Castanea mollissima* variety, with a

higher average single nut weight of 25 g. "Kuili" showed a higher leaf length and width, an elliptic blade, and average single blade area of ~120 cm$^2$, with a higher total chlorophyll content and photosynthesis capacity. "YLZ1" is a variety of *Castanea henryi*; the single weight of its nuts is relatively small, with an average single nut weight of 7~8 g. Its leaves are small and narrow, and the average leaf area is ~60 cm$^2$, with a lower total chlorophyll content and photosynthesis capacity. However, the nut quality of "YLZ1" is higher than most *Castanea mollissima* varieties, with a starch content of 67.2% and a higher total soluble sugar content of 13.8%. The $F_1$ cross between 'Kuili' and 'YLZ1' over two successive years was chosen for further analysis.

The $F_1$ mapping population consisted of 200 progenies generated by crossing 'Kuili' and 'YLZ1'. The hybrid seed was obtained in the year of 2013 through artificial pollination, sown, and seedlings were obtained in the year of 2014. In the spring of 2015, the $F_1$ progeny seedlings were planted in the 'Experimental Forest Farm of Qingyuan' (Qingyuan County, Lishui, China). The hybrid progeny seedlings were planted with a spacing of 4 m × 4 m. For the measurements, four-year-old seedlings with uniform height and healthy leaves were selected. Young leaves were harvested from an F1 population of 183 individuals along with the parents, immediately frozen in liquid nitrogen, and then transferred to the laboratory and stored in a −80 °C refrigerator. The DNA was extracted from leaf samples using the improved CTAB method [11]. A NanoDrop-1000 spectrophotometer (Nano Drop, Wilmington, DE, USA) was used for the quantification of the DNA concentration, which was finally adjusted to a concentration of 50 ng/μL. We confirmed the double-stranded DNA with a concentration of at least 100 ng/μL using a Qubit 3.0 fluorescence analyzer (Thermo Fisher Scientific, Waltham, MA, USA).

## 2.2. Sequencing Library Construction and Illumina Sequencing

A GBS pre-design experiment was performed for evaluation of the enzymes as well as the sizes of the restriction fragments, which obey three criteria, to improve the efficiency of GBS [12,19]. Moreover, fragments with a length range of ~50 bp were used to maintain uniformity in the depth of the different fragments. Second, the GBS library was constructed according to the instructions described previously [12]. After that, the paired-end sequencing of the selected tags were calculated and analyzed on an lllumina high-throughput sequencing platform (https://cn.novogene.com/, 10 August 2019), and then the SNP genotyping and evaluation were carried out.

## 2.3. Quality Assessment

The sorted samples sequences were based on their barcodes. A series of in-house C scripts for quality control (QC) were applied for the raw data in order to ensure the reads used were reliable and can be used. The unqualified types of reads will be filtered using QC procedures, which were performed according to the previous description [12]. The initial data underwent a rigorous filtering process to ensure data quality and reliability. Reads containing adapter or junction sequences were meticulously excluded from the dataset. To maintain data integrity, paired reads were carefully removed if the proportion of 'N' bases in the single-end sequencing read exceeded 10% of the read length. Additionally, to enhance data accuracy, paired reads were diligently discarded if the proportion of low-quality bases (with a quality value of Q $\leq$ 5) in the single-end sequencing read exceeded 50% of the read length. For the filtering process, crucial parameter settings were implemented using vcftools software (v 0.1.16, https://vcftools.sourceforge.net/, 10 June 2020); the minor allele frequency (maf) was set to 0.05, ensuring that rare variants with a frequency below 5% were excluded from the analysis. The missing data threshold (missing) was set to 0.9, allowing only markers with a missing data rate below 10% to be retained in the final dataset.

## 2.4. SNP Identification and Genotyping

We first compared the clean reads of each sample by referring to the Chinese chestnut genome (http://gigadb.org/dataset/view/id/100643, accessed on 15 August 2019), using

the Burrows-Wheeler Aligner (BWA) software (version: 0.7.17, http://bio-bwa.sourceforge.net, 3 May 2021) [26]. SAMtools software was used to convert alignment files into bam files (version: 0.1.17, http://samtools.sourceforge.net, 15 May 2021) [26]. The read pairs with the highest mapping quality was retained, where the multiple read pairs had identical external coordinates. GATK software (version: 4.1.9.0) was used to assign variable calls to all samples. Then, Perl scripts were used to filter the SNPs and ANNOVAR (http://www.openbioinformatics.org/annovar/, 10 June 2022) used to annotate the SNPs with GFF3 files from the reference Chinese chestnut genome [27]. Eight separation patterns (aa × bb, lm × ll, nn × np, ab × cc, hk × hk, cc × ab, ef × eg and ab × cd) were classified using the SNP markers. The segregation patterns <nn × np>, <lm × ll> and <hk × hk> were used to construct the gene map in the $F_1$ population. The polymorphic heterozygous SNP markers found in only one parent and in both parents were denoted as <nn × np> or <lm × ll>, and the heterozygous markers found in both parents were denoted as <hk × hk>. A chi-square test was performed for the separation ratio of markers, and the markers ($p > 0.001$) that conform to the expected Mendelian separation ratio can be included in this mapping.

### 2.5. Genetic Map Construction

The genetic map construction was performed using JoinMapR 4.0 software [28]. Also, we excluded markers with segregation distortions ($p < 0.001$), missing more than 5% of data, or containing abnormal bases. Then, we clustered the filtered markers with logarithm-of-odds (LOD) values on different gradients. The results showed that when the LOD = 2, we achieved ideal clustering results. The cluster result showed 12 major linkage groups, as well as good correspondence with the chromosome information of the genome. Regression-based parameters were used to construct maternal, paternal and integrated maps. Meanwhile, the Kosambi algorithm was used to classify each group's markers and calculate genetic distance [29].

### 2.6. Phenotypic Data Analysis

Leaves were selected from the F1 population, along with the two parents, during maturity. The leaf traits included chl a + b, chla, chlb, chla/b, LA, LDW, LFW, LMC, LL, LW, PL, LSI, SLW, Gs, Ci, Pn, Tr and WUE. chla and chlb were detected, as previously described [30]. chla/b is the ratio of the values of chla and chlb. Pn, Tr, Gs and Ci were measured using a portable photosynthesis system (Li6400; LICOR, Huntington Beach, CA, USA); the parameters were set as previously described [30]. The WUE was calculated using the Pn/Tr ratio. LMC and SLW were performed following previously described methods [31]. At least 15 leaves and 3 repeats from each tree were used for the detection of the leaf traits data. The average values of each trait per individual were used for QTL analysis.

### 2.7. Analysis of QTLs

QTL mapping of the 18 leaf traits was accomplished using the composite interval mapping (CIM) model of MapQTL 6.0 [32]. The composite interval mapping (CIM) model effectively removes the influence of genetic loci outside the current intervals. Additionally, it prevents the current detection interval from being affected by other interval quantitative trait loci (QTL) present in the simple interval mapping (SIM) model. Moreover, CIM proves to be more applicable in scenarios where multiple QTLs exist on a single chromosome within the map, such as Lg 2 and 7. As a result, compared to the SIM model, CIM significantly enhances the accuracy of the mapping process. To determine the LOD threshold, each trait was calculated separately via 1000 permutations ($P = 0.05$). QTLs with LODs above the threshold were screened and considered significant. The calculation of phenotypic variance (PVE) for a single QTL was based on the maximum likelihood estimated; the dominant QTL must be PVE $\geq$ 15% in order to be estimated. After screening, the Perl SVG module was used for the QTLs that were drawn on LGs.

### 2.8. Statistical Analysis

SPSS statistical software (version 19.0) was used to analyze the phenotypic data, as well as the frequency distribution. The frequency distribution normality was estimated via the kurtosis and skewness [33], and two-tailed bivariate correlation tests were used to analyze the correlations among the chl a + b, chla, chlb, chla/b, LA, LDW, LFW, LMC, LL, LW, PL, LSI, SLW, Gs, Ci, Pn, Tr, and WUE traits.

## 3. Results

### 3.1. Genotyping by Sequencing

We found that the average mapping rate of parents and F1 individuals was 97.46%, with an average depth of 15.51%, and 96.8%, with an average depth of 11.05%, respectively. High-quality raw sequencing data were ~116.31 G. After data trimming and filtering, the generated data that met the requirement of high quality, for the parents, were 26 G, with an average of 13 G. The screened quality data generated for the offspring were 94.28 G, with an average of 0.52 G. The Q30 ratio was 92.32%, and the content of guanine cytosine (GC) was 36.54%. We obtained a total of 515,193,398 clean reads from 183 individuals, with 15,151,546,800 and 10,817,349,600 clean reads in the maternal parent 'Kuili' and the paternal parent 'YLZ1', respectively. After the filtering, the clean reads from 183 samples ranged from 0.2 G to 1 G, with an average of 0.52 G, and 96.8% of clean reads successfully matched the Chinese chestnut genome. Based on sequencing coverage and average sequencing depth, the F1 individual $1 \times$ coverage rate was 7%~17.87%, the $4 \times$ coverage rate was 2.36%~7.79%, and the average sequencing depth was 11.05%.

### 3.2. Linkage Map Construction

We utilized JoinMap® 4.0 to construct three high-density genetic maps from the cross of 'Kuili' and 'YLZ1'. We obtained a total of 4578 SNP markers, which were assigned to 12 LGs (LG01-LG12) with a length of 1812.46 cM. The average distance between adjacent markers in the integrated map was 0.4 cM (Figure 1, Tables 1 and S1). The average distance between adjacent markers in each LG was 0.27 cM (LG 11) to 1.15 cM (LG 9) (Table 1). The male parent 'YLZ1' contains 3270 SNP markers in 12 LGs, and the map length was 2168.7 cM, with an average distance of 0.66 cM between adjacent markers (Table S2). However, the average distance between adjacent markers ranged from 0.48 cM (LG 3, 5, and 11) to 2.1 cM (LG 10) (Table S2). The female parent 'Kuili' consists of 2230 markers from 12 LGs, with a map length of 1319.58 cM and an average interval of 0.59 cM between adjacent markers ranging from 0.37 cM (LG12) to 1.67 cM (LG9). The longest was LG6 (297.42 cM), and the shortest was LG11 (53.36 cM) among the three maps (Table S3).

**Table 1.** Analysis of the SNP markers' distribution in the integrated genetic linkage map.

| Linkage Group | Average Distance between Two Markers (cM) | Number of Intervals-D (cM) | | | |
|---|---|---|---|---|---|
| | | <5 | 5~10 | 10~20 | >20 |
| Lg01 | 0.31 | 315 | 2 | 0 | 0 |
| Lg02 | 0.42 | 487 | 0 | 0 | 0 |
| Lg03 | 0.29 | 584 | 0 | 0 | 0 |
| Lg04 | 0.28 | 485 | 0 | 0 | 0 |
| Lg05 | 0.3 | 338 | 1 | 0 | 0 |
| Lg06 | 0.36 | 676 | 3 | 0 | 0 |
| Lg07 | 0.33 | 367 | 0 | 0 | 0 |
| Lg08 | 0.34 | 310 | 0 | 0 | 0 |
| Lg09 | 1.15 | 164 | 7 | 0 | 0 |
| Lg10 | 1.01 | 206 | 8 | 0 | 0 |
| Lg11 | 0.27 | 343 | 0 | 0 | 0 |
| Lg12 | 0.45 | 279 | 3 | 0 | 0 |
| Total | 0.4 | 4554 | 24 | 0 | 0 |

D, distance between adjacent markers.

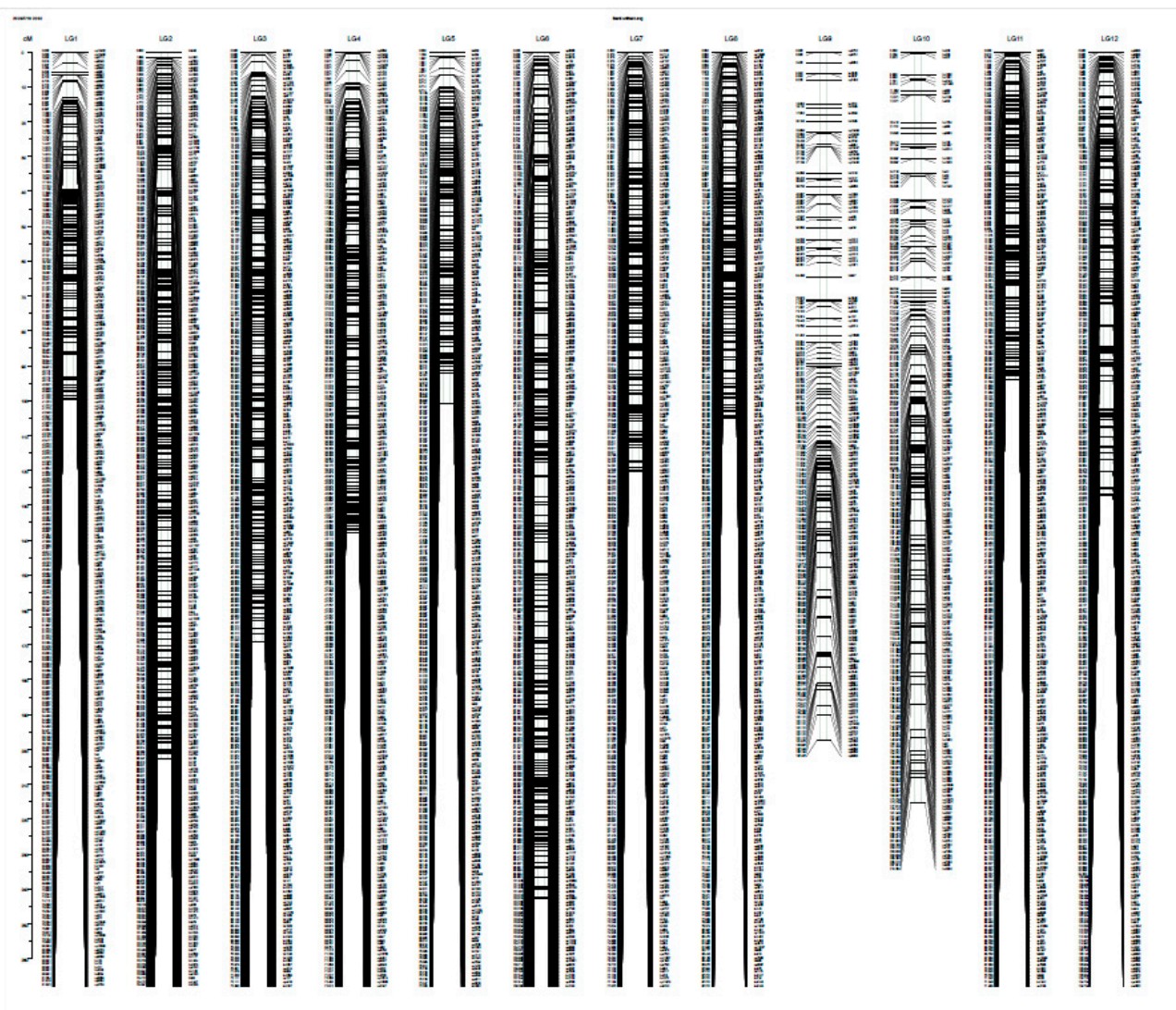

**Figure 1.** Integrated LGs in the Chinese chestnut genome using the 'Kuili' and 'YLZ1' cross.

The results showed that the maximum number of markers for 'YLZ1' in LG 6 was 297.42, the maximum number of markers for 'Kuili' in LG 10 was 251.82, and the maximum number of markers for the integrated map in LG 6 was 242.76. The minimum number of markers for 'YLZ1' in LG 5 was 108.84, the minimum number of markers of LG 11 for 'Kuili' and the integrated map was 53.36 and 93.87, respectively. As shown in Table 1, 12 LGs covered most of the SNP markers. There was a total of 4578 intervals between adjacent markers among the 12 LGs, the number of intervals less than 5 cM was 4554, and the number of intervals within 5 cM < D < 10 cM was 24 (Table 1).

*3.3. Phenotypic Analysis*

We found wide segregation and variation in the $F_1$ population's leaf traits (Table S4). We calculated the data of the leaf traits, and presented these as the mean, SD, CV, skewness and kurtosis (Table S4); all traits were normally distributed in the two consecutive years studied. As shown in Table 2, Chl a, Chl b and Chl a + b were highly significantly correlated with LL, LW, LA, PL, LFW, LDW, Pn and Gs. Pn, Gs and Tr showed a positive significant correlation with LL, LW, LA, PL, LFW and LDW (Table 2). Pn and Gs also showed a positive significant correlation with SLW (Table 2). Furthermore, we found a significant negative correlation between WUE and Ci or Tr, with correlation coefficients of −0.254 ** and −0.532 **, respectively (Table 2).

**Table 2.** Pearson correlation analysis of leaf phenotypic traits and photosynthetic physiological indices in $F_1$ population.

| Trait/Trait | LL | LW | LSI | LA | PL | LFW | LDW | LMC | SLW | $P_n$ | $G_s$ | $C_i$ | $T_r$ | WUE | Chl·a | Chl·b | Chl·a + b | Chl·a/b |
|---|---|---|---|---|---|---|---|---|---|---|---|---|---|---|---|---|---|---|
| LL | 1.000 | | | | | | | | | | | | | | | | | |
| LW | 0.728 ** | 1.000 | | | | | | | | | | | | | | | | |
| LSI | 0.410 ** | −0.322 ** | 1.000 | | | | | | | | | | | | | | | |
| LA | 0.899 ** | 0.940 ** | −0.009 | 1.000 | | | | | | | | | | | | | | |
| PL | 0.640 ** | 0.442 ** | 0.289 ** | 0.573 ** | 1.000 | | | | | | | | | | | | | |
| LFW | 0.814 ** | 0.832 ** | 0.024 | 0.904 ** | 0.657 ** | 1.000 | | | | | | | | | | | | |
| LDW | 0.799 ** | 0.808 ** | 0.032 | 0.882 ** | 0.689 ** | 0.959 ** | 1.000 | | | | | | | | | | | |
| LMC | 0.284 ** | 0.288 ** | 0.028 | 0.308 ** | 0.019 | 0.285 ** | 0.070 | 1.000 | | | | | | | | | | |
| SLW | 0.170 * | 0.113 | 0.093 | 0.165 * | 0.484 ** | 0.476 ** | 0.601 ** | −0.405 ** | 1.000 | | | | | | | | | |
| $P_n$ | 0.342 ** | 0.325 ** | 0.041 | 0.352 ** | 0.372 ** | 0.431 ** | 0.429 ** | 0.065 | 0.308 ** | 1.000 | | | | | | | | |
| $G_s$ | 0.316 ** | 0.289 ** | 0.052 | 0.325 ** | 0.361 ** | 0.419 ** | 0.400 ** | 0.068 | 0.279 ** | 0.825 ** | 1.000 | | | | | | | |
| $C_i$ | −0.062 | −0.043 | −0.023 | −0.059 | −0.086 | −0.022 | −0.059 | 0.124 | −0.034 | 0.188 * | 0.521 ** | 1.000 | | | | | | |
| $T_r$ | 0.369 ** | 0.421 ** | −0.050 | 0.431 ** | 0.263 ** | 0.474 ** | 0.427 ** | 0.202 ** | 0.175 * | 0.616 ** | 0.723 ** | 0.391 ** | 1.000 | | | | | |
| WUE | −0.105 | −0.179 * | 0.096 | −0.166 * | 0.049 | −0.122 | −0.067 | −0.184 * | 0.137 | 0.301 ** | 0.004 | −0.254 ** | −0.532 ** | 1.000 | | | | |
| Chl·a | 0.359 ** | 0.316 ** | 0.078 | 0.358 ** | 0.297 ** | 0.291 ** | 0.296 ** | 0.052 | 0.017 | 0.327 ** | 0.283 ** | 0.029 | 0.226 ** | 0.049 | 1.000 | | | |
| Chl·b | 0.349 ** | 0.305 ** | 0.080 | 0.341 ** | 0.275 ** | 0.276 ** | 0.276 ** | 0.064 | 0.001 | 0.312 ** | 0.278 ** | 0.064 | 0.190 * | 0.069 | 0.965 ** | 1.000 | | |
| Chl·a + b | 0.360 ** | 0.321 ** | 0.074 | 0.360 ** | 0.295 ** | 0.291 ** | 0.295 ** | 0.062 | 0.009 | 0.330 ** | 0.283 ** | 0.037 | 0.223 ** | 0.052 | 0.997 ** | 0.979 ** | 1.000 | |
| Chl·a/b | −0.002 | −0.017 | 0.013 | 0.011 | 0.052 | 0.007 | 0.032 | −0.088 | 0.060 | 0.023 | −0.014 | −0.108 | 0.113 | −0.072 | 0.010 | −0.240 ** | −0.050 | 1.000 |

Note: *, ** showed significantly correlations at 0.05 and 0.01 levels, respectively.

### 3.4. QTLs for Leaf Traits

The results showed that 71 QTLs are located on lg 2, 3, 6, 7, and 12, according to the constructed integrated genetic map depicting the leaf traits of LA, LFW, LL, LW, Gs, PL, LDW, chlb and SLW (Figure 2, Table 3). The phenotypic variance of the Chinese chestnut leaf ranged from 10 to 18.8%, and the LOD values of leaf traits ranged from 4.19 to 8.28%. There were 15 QTLs for LA, 1 for LDW, and 13 for LW, which were major QTLs.

**Table 3.** Quantitative trait loci analysis of leaf traits in the $F_1$ population.

| Traits | lg | QTLs | Position (cM) | Marker | LOD | PVE (%) |
|--------|----|------|---------------|--------|-----|---------|
| chlb | 7 | qchlb-7-1 | 43.967 | lm1378 | 4.4 | 10.5 |
| | 7 | qchlb-7-2 | 27.848 | hk1061 | 4.37 | 10.4 |
| | 7 | qchlb-7-3 | 31.841 | hk727 | 4.36 | 10.4 |
| | 7 | qchlb-7-5 | 34.207 | np2184 | 4.35 | 10.4 |
| | 7 | qchlb-7-6 | 77.799 | lm2630 | 4.33 | 10.3 |
| | 7 | qchlb-7-7 | 78.076 | np2214 | 4.32 | 10.3 |
| | 7 | qchlb-7-8 | 55.739 | lm2145 | 4.29 | 10.2 |
| | 7 | qchlb-7-10 | 34.207 | np3304 | 4.28 | 10.2 |
| | 7 | qchlb-7-11 | 27.848 | lm2 | 4.26 | 10.2 |
| | 7 | qchlb-7-12 | 36.824 | hk725 | 4.19 | 10 |
| Gs | 3 | qGs-3-1 | 19.122 | np10917 | 4.37 | 10.4 |
| | 3 | qGs-3-2 | 19.131 | np1854 | 4.33 | 10.3 |
| | 3 | qGs-3-3 | 19.131 | np1855 | 4.29 | 10.2 |
| LA | 2 | qLA-2-1 | 101.906 | hk1025 | 6.99 | 16.1 |
| | 2 | qLA-2-2 | 109.674 | hk189 | 6.89 | 15.9 |
| | 2 | qLA-2-3 | 110.502 | hk1261 | 6.75 | 15.6 |
| | 2 | qLA-2-4 | 110.034 | lm574 | 6.68 | 15.5 |
| | 2 | qLA-2-5 | 101.563 | lm2215 | 6.63 | 15.4 |
| | 2 | qLA-2-6 | 111.088 | lm2450 | 6.57 | 15.2 |
| | 2 | qLA-2-7 | 115.47 | hk1267 | 6.5 | 15.1 |
| | 2 | qLA-2-8 | 115.501 | lm2373 | 6.47 | 15 |
| | 2 | qLA-2-9 | 102.18 | hk1203 | 6.39 | 14.9 |
| | 2 | qLA-2-10 | 115.501 | lm2293 | 6.36 | 14.8 |
| | 2 | qLA-2-11 | 111.333 | np10396 | 6.33 | 14.7 |
| | 2 | qLA-2-12 | 102.318 | lm1661 | 6.28 | 14.6 |
| | 2 | qLA-2-13 | 115.439 | lm3035 | 6.11 | 14.2 |
| | 2 | qLA-2-14 | 101.906 | hk1030 | 6.09 | 14.2 |
| | 2 | qLA-2-15 | 111.811 | hk1266 | 6.07 | 14.2 |
| LDW | 6 | qLDW-6-1 | 89.042 | hk1386 | 6.47 | 15 |
| | 6 | qLDW-6-2 | 210.151 | hk155 | 5.78 | 13.5 |
| | 6 | qLDW-6-3 | 212.351 | lm1676 | 5.77 | 13.5 |
| | 6 | qLDW-6-4 | 89.042 | lm2681 | 5.62 | 13.2 |
| | 6 | qLDW-6-5 | 216.417 | lm2824 | 5.59 | 13.1 |
| | 6 | qLDW-6-6 | 212.351 | lm438 | 5.58 | 13.1 |
| | 6 | qLDW-6-8 | 213.178 | hk1433 | 5.54 | 13 |
| | 6 | qLDW-6-9 | 74.256 | np4288 | 5.52 | 13 |
| | 6 | qLDW-6-10 | 210.151 | lm1018 | 5.5 | 12.9 |
| LFW | 2 | qLFW-2-1 | 109.674 | hk189 | 6.19 | 14.4 |
| | 2 | qLFW-2-2 | 110.502 | hk1261 | 5.81 | 13.6 |
| | 2 | qLFW-2-3 | 110.034 | lm574 | 5.77 | 13.5 |
| | 2 | qLFW-2-4 | 89.042 | hk1386 | 5.77 | 13.5 |
| | 2 | qLFW-2-5 | 101.906 | hk1025 | 5.74 | 13.4 |
| | 2 | qLFW-2-6 | 101.563 | lm2215 | 5.67 | 13.3 |
| | 2 | qLFW-2-7 | 115.47 | hk1267 | 5.62 | 13.2 |
| | 2 | qLFW-2-8 | 115.501 | lm2373 | 5.61 | 13.2 |
| LL | 2 | qLL-2-1 | 109.674 | hk189 | 5.59 | 13.1 |
| | 2 | qLL-2-2 | 110.502 | hk1261 | 5.51 | 12.9 |
| | 2 | qLL-2-3 | 110.034 | lm574 | 5.35 | 12.6 |
| | 2 | qLL-2-4 | 115.501 | lm2373 | 5.35 | 12.6 |

**Table 3.** *Cont*.

| Traits | lg | QTLs | Position (cM) | Marker | LOD | PVE (%) |
|---|---|---|---|---|---|---|
| | 2 | qLL-2-5 | 115.47 | hk1267 | 5.33 | 12.6 |
| | 2 | qLL-2-6 | 101.906 | hk1025 | 5.28 | 12.4 |
| | 2 | qLL-2-7 | 101.563 | lm2215 | 5.27 | 12.4 |
| | 2 | qLL-2-8 | 111.088 | lm2450 | 5.26 | 12.4 |
| LW | 2 | qLW-2-1 | 101.906 | hk1025 | 8.28 | 18.8 |
| | 2 | qLW-2-2 | 102.18 | hk1203 | 7.81 | 17.8 |
| | 2 | qLW-2-3 | 111.088 | lm2450 | 7.59 | 17.4 |
| | 2 | qLW-2-4 | 101.563 | lm2215 | 7.57 | 17.4 |
| | 2 | qLW-2-5 | 109.674 | hk189 | 7.47 | 17.1 |
| | 2 | qLW-2-6 | 115.47 | hk1267 | 7.46 | 17.1 |
| | 2 | qLW-2-7 | 110.502 | hk1261 | 7.45 | 17.1 |
| | 2 | qLW-2-8 | 102.318 | lm1661 | 7.4 | 17 |
| | 2 | qLW-2-9 | 111.333 | np10396 | 7.38 | 17 |
| | 2 | qLW-2-10 | 115.501 | lm2373 | 7.35 | 16.9 |
| | 2 | qLW-2-11 | 115.501 | lm2293 | 7.31 | 16.8 |
| | 2 | qLW-2-12 | 110.034 | lm574 | 7.28 | 16.7 |
| | 2 | qLW-2-13 | 101.906 | hk1030 | 6.99 | 16.1 |
| PL | 3 | qPL-3-1 | 49.107 | lm897 | 5.55 | 13 |
| | 3 | qPL-3-2 | 48.833 | hk898 | 5.54 | 13 |
| | 3 | qPL-3-3 | 49.107 | hk472 | 5.4 | 12.7 |
| | 3 | qPL-3-4 | 130.784 | hk763 | 5.1 | 12 |
| SLW | 12 | qSLW-12-1 | 18.321 | hk1366 | 4.47 | 10.6 |

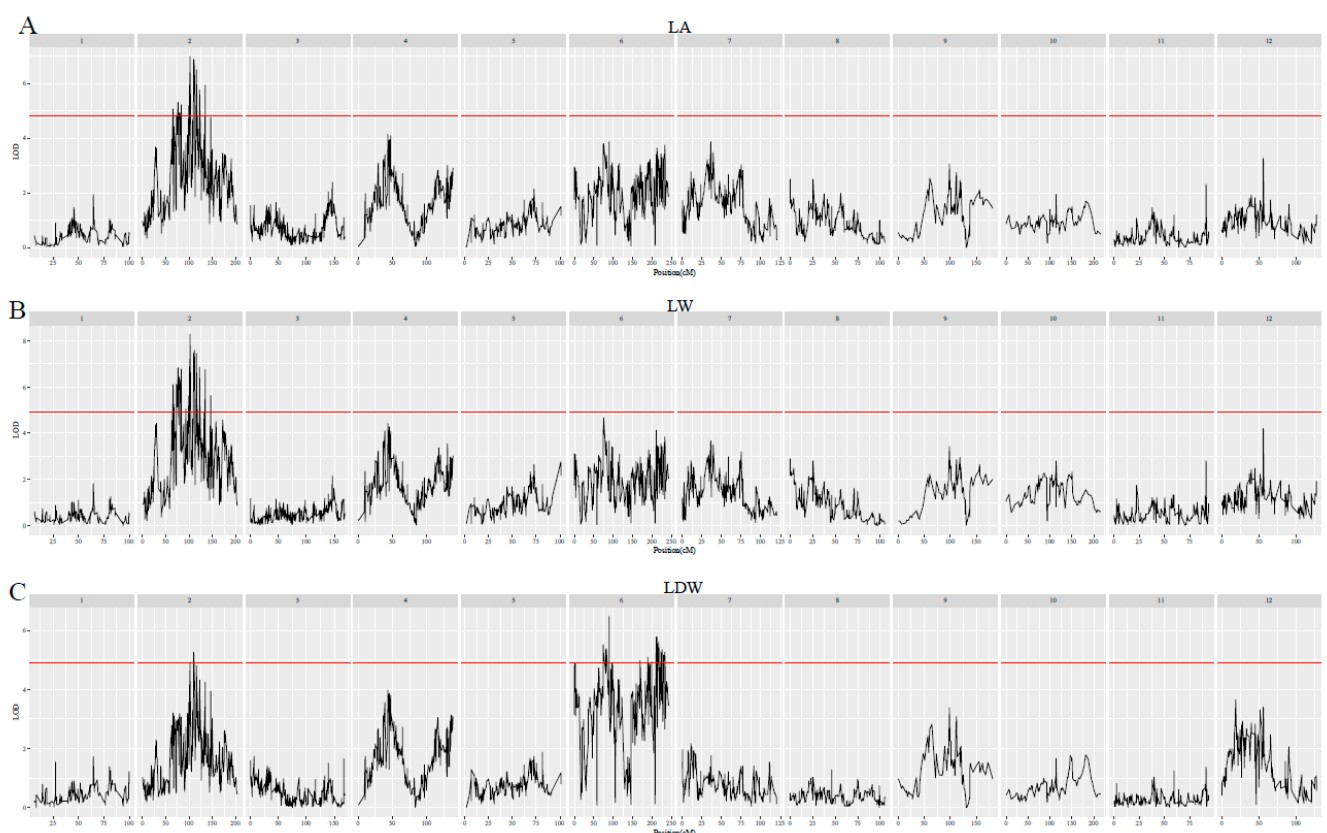

**Figure 2.** *Cont*.

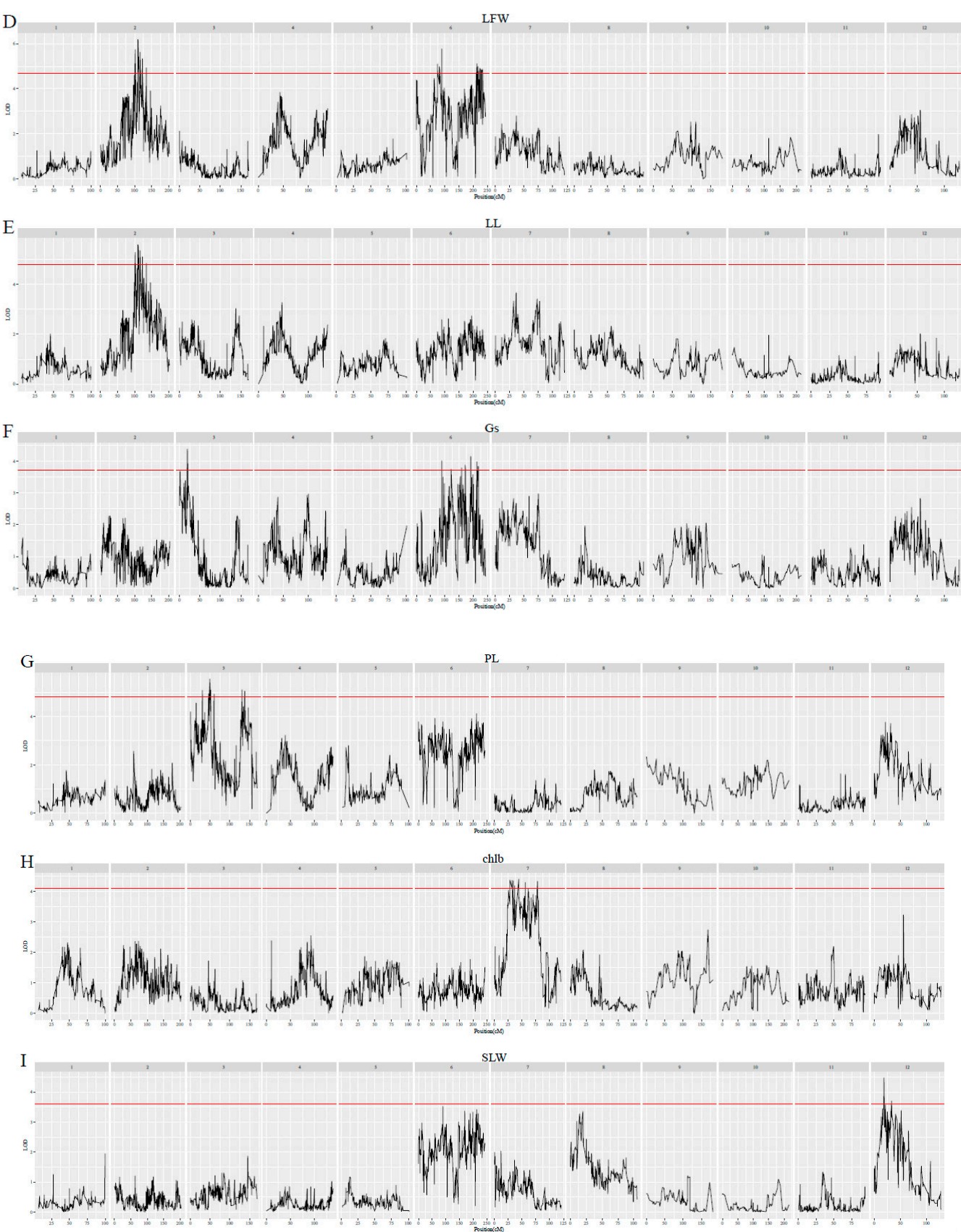

**Figure 2.** Distribution of genetic markers on the LGs of LA (**A**), LW (**B**), LDW (**C**), LFW (**D**), LL (**E**), Gs (**F**), PL (**G**), chlb (**H**), and SLW (**I**).

Fifteen QTLs were identified for LA, including qLA-2-1, qLA-2-2, qLA-2-3, qLA-2-4, qLA-2-5, qLA-2-6, qLA-2-7 and qLA-2-8. qLA-2-1 was present in lg 2, in a region centered at 101.91 cM, with a phenotypic variance of 16.1% (Table 3). Seven QTLs were detected, including qLA-2-2, qLA-2-3, qLA-2-4, qLA-2-5, qLA-2-6, qLA-2-7 and *qLA-2-8* in lg 2 in a region centered between at 101.563~115.501 cM, with a phenotypic variance $\geq$ 15%.

Thirteen QTLs were identified for LW; *qLW-2-1* was located on lg 2, in the central region at 101.906 cM, with a phenotypic variance of 18.8% (Table 3). Eight QTLs, including qLW-2-2, qLW-2-3, qLW-2-4, qLW-2-5, qLW-2-6, qLW-2-7, qLW-2-8 and qLW-2-9, identified for LW, were in a region between 101.563 and 115.47 cM, with a phenotypic variance from 17%~17.8% (Table 3). Moreover, the other four QTLs, i.e., qLW-2-10, qLW-2-11, qLW-2-12, qLW-2-13 for LW explained 16.1 to 16.9% of the phenotypic variance (Table 3).

One QTL was identified for LDW, and qLDW6-1 located on lg 6 was at the center of the region of 89.04 cM, with a phenotypic variance of 15%. Moreover, QTLs located on lg 2 also control the LA, LFW, LL, and LW leaf traits (Table 3).

## 4. Discussion

The Chinese chestnut (*Castanea* Mill.) holds the distinction of being the most widely distributed species in China, boasting an incredibly diverse array of germplasm resources. Notably, *C. henryi* thrives primarily in the mountainous areas of southern Zhejiang and northern Fujian. Its nuts are highly esteemed for their elevated sugar content, waxy texture, and exceptional quality, making them exceptionally suitable for various processing applications. However, in comparison to *C. mollissima*, dominant varieties of *C. henryi* with large nuts, high yields, and strong environmental adaptability are relatively scarce. This scarcity has hindered the comprehensive development of chestnut cultivation [11,12]. Thus, it becomes of utmost importance to harness the potential of interspecific remote hybridization and gene recombination technologies, capitalizing on the advantageous traits present in both southern *C. mollissima* and *C. henryi*. Through this strategic approach, we can carefully select and breed new variants that showcase outstanding nut traits, constituting a vital step in the advancement of the Chinese chestnut industry.

Numerous studies have been dedicated to intraspecific hybridization of the Chinese chestnut, as well as interspecific hybridization between the Chinese chestnut and the Japanese chestnut, in the pursuit of breeding advancements [12,34–36]. However, research on interspecific crossbreeding between *C. mollissima* and *C. henryi* remains limited. It is essential to note that *Castanea* species are perennial woody plants, characterized by a prolonged juvenile phase, which extends the time required to obtain nuts for evaluation, usually taking around 6 to 7 years. This elongated breeding cycle presents challenges in the breeding process. To overcome these challenges and expedite the breeding process, researchers have proposed an efficient approach. This involves constructing a high-density genetic linkage map of chestnut, establishing the relationships between specific traits and molecular markers. In doing so, it becomes possible to determine the number of genes responsible for the target traits, understand their effect values, and identify potential interactions between genes. Implementing a molecular-assisted breeding system based on this genetic knowledge can significantly accelerate the development of new chestnut varieties, ultimately reducing the time needed to produce improved cultivars.

### 4.1. Construction of a High-Density Genetic Map of Chestnut

Multiple genetic maps of the *Castanea* genus have been developed, each revealing valuable quantitative trait loci (QTLs) associated with growth, fruit quality, and disease resistance. In the case of the Chinese chestnut, a genetic map was constructed using the $F_2$ population resulting from interspecific hybridization between American chestnut and Chinese chestnut. By utilizing this map, a three-QTL model was established to account for approximately 70% of the phenotypic variation of chestnut blight disease [37]. Another genetic map was created for *Castanea sativa*, using a diverse population of 152 $F_1$ individuals resulting from the crossing of two Turkish chestnut populations. This map integrated

142 RAPDs, 3 isozymes, 30 ISSRs, and 42 SSRs markers, spanning a genetic distance of 832.9 cM [22]. This map played a key role in the identification of 35 phenological QTLs, 28 growth QTLs, and 17 carbon isotope QTLs [22]. Moreover, researchers utilized the genetic map of the entire *Castanea* genus to discover 329 SSRs and 1064 SNP markers. These markers were obtained by combining expressed sequence tag (EST) data with the physical maps of the genome [36]. This approach allowed for a more comprehensive understanding of the genetic diversity and marker distribution within the genus.

So far, a total of nine genetic maps have been produced for the study of *Castanea*. However, apart from the genetic map described by Ji et al. (2018), they have not succeeded in improving the genetic map's density using RAPD, RFLP, and ISSR markers. Recently, a new genetic map for chestnut was constructed, comprising 2620 SNP markers with an average marker distance of 0.41 cM. This map was developed using 259 $F_1$ populations, and proved useful in identifying QTLs associated with nut thickness, maturity, single nut weight, and nut width [12]. In our study, we were able to map the largest number of SNP markers (4578 SNP markers) using GBS sequencing. This number significantly surpassed the marker count obtained in previous maps [12,21,22,36,37]. Meanwhile, our map, with a marker interval of 0.4 cM, showed the shortest average genetic interval and the highest density compared to any previous maps constructed for this genus. Additionally, a novel genetic map for *Castanea henryi*, named 'YLZ1', was created, significantly expanding the genomic resources available for chestnut breeding (Figure 3). It is worth noting that Chinese chestnut was chosen as the mapping parent for seven out of the ten existing *Castanea* genetic maps. This selection was based on its desirable characteristics, including high resistance to stress, superior nut quality, and strong adaptability to various environmental conditions. These genetic resources and tools hold great potential for ongoing and future chestnut breeding efforts.

### 4.2. Analysis of QTLs for Leaf Traits

Some leaf traits are related to stress resistance, for example, resistance to cold and drought [38]. In this study, we first identified 71 QTLs related to the leaf traits of Chinese chestnut. These leaf traits, including Chlb, Gs, LDW, LFW, LL, LW, PL and SLW, are closely related to the growth and stress resistance of Chinese chestnut. Leaf length, width, thickness, area are affected by genotype and environment [39–42]. Leaf shape and number also important traits that determine a plant's appearance, photosynthesis capacity, light energy utilization, yield, water WUE, and even desirability to consumers [43–45]. Fifteen QTLs for LA, one QTL for LDW, and thirteen QTLs for LW were major QTLs identified in this study that control leaf traits which then affect the physiology of Chinese chestnut. Leaf traits also affect nitrogen and phosphorus resorption, which reflect a plant's nutrient conservation strategy [46]. More shade-tolerant plant species showed higher leaf K contents, thickness, *Fv/Fm* values and qN values, which enhance their resistance to light stress [47]. In addition, we identified ten QTLs for chlb and three QTLs for Gs, which might affect important leaf traits that play a part in photosynthesis and light stress.

The quantitative trait loci of leaf traits provide theoretical support for breeding programs, and many quantitative trait loci of crops have already been found and applied in breeding engineering [44,48]. Siddique et al. (2021) constructed a linkage map and identified seven major and minor effects of QTLs, and candidate genes from QTLs were related to the photosynthesis and chlorophyll content of strawberry leaves. Bu et al. (2022) used single-segment substitution lines to identify five QTLs associated with leaf number, and the QTLs expressed their effects mainly in three growth stages. Based on the linkage map of 'Lady's Slipper Orchids' twelve QTLs were identified, being linked to leaf length, leaf width, leaf thickness, and leaf number [49]. In the present study, we identified fifteen QTLs for LA, nine QTLs for LDW, eight QTLs for LFW, eight QTLs for LL, thirteen QTLs for LW, four QTLs for PL and one QTL for SLW. These leaf traits provide useful theoretical indications of high nut quality, which may inform the production of a Chinese chestnut

breeding program. By constructing a genetic linkage map, QTLs for other leaf traits (such as anthocyanin and shade-avoiding syndrome) can also be identified [50–52].

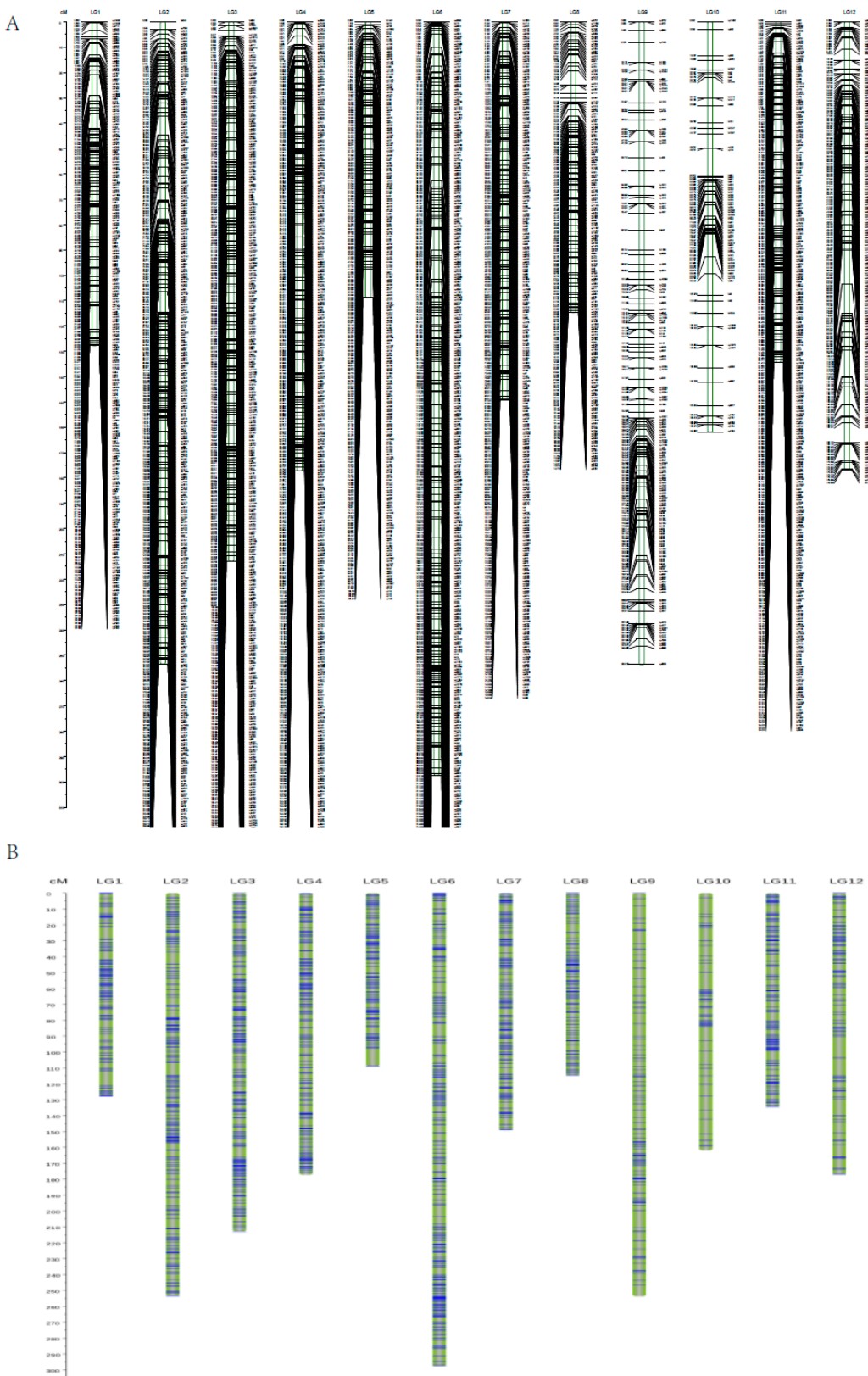

**Figure 3.** LGs in the Chinese chestnut using the 'YLZ1'. Two display forms of LGs of 'YLZ1' (**A**,**B**).

## 5. Conclusions

In sum, we constructed a linkage map of chestnut leaf traits with high density and the shortest average genetic interval, using identified SNP markers obtained via GBS sequencing. The genetic linkage map contains 4578 SNP markers, covering an area of 1812.46 cM, with an average distance of 0.4 cM between adjacent markers. Based on this genetic map, a total of 71 QTLs for nine leaf traits were identified, including ten chl b QTLs, three Gs QTLs, fifteen LA QTLs, nine LDW QTLs, eight LFW QTLs, eight LL QTLs, thirteen LW QTLs, four PL QTLs, and one SLW QTL. Our findings might helpful for producing high yields through Chinese chestnut breeding practices.

**Supplementary Materials:** The following supporting information can be downloaded at: https://www.mdpi.com/article/10.3390/f14081684/s1, Table S1. Details of the genetic linkage maps from the crossing of 'Kuili' and 'YLZ1'; Table S2. Details of the genetic linkage map of male parent 'YLZ1'; Table S3. Details of the genetic linkage map of female parent 'Kuili'; Table S4. Population segregation characteristics of leaf phenotypic traits and photosynthetic physiological indices in $F_1$ generation.

**Author Contributions:** Conceptualization, X.J. and B.G.; Methodology, X.J., Y.W. and J.L.; Software, J.W. and W.H.; Formal Analysis, Y.W., X.J. and W.H.; Investigation, X.J., X.W. and C.W.; Resources, X.J. and B.G.; Data Curation, Y.W.; Writing—Original Draft Preparation, X.J.; Writing—Review and Editing, X.J., B.G. and Y.W.; Visualization, X.J.; Supervision, Y.W. and B.G.; Project Administration, X.J. and B.G.; Funding Acquisition, X.J. and B.G. All authors have read and agreed to the published version of the manuscript.

**Funding:** This research was supported by the Zhejiang Science and Technology Major Program on Agriculture New Variety Breeding (2021C02070-4).

**Institutional Review Board Statement:** Not applicable.

**Informed Consent Statement:** Not applicable.

**Data Availability Statement:** The data presented in this study are available on request from the corresponding author.

**Conflicts of Interest:** The authors declare no conflict of interest.

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
