# Peer review of "The Construction of a High-Density Genetic Map for the Interspecific Cross of Castanea mollissima × C. henryi and the Identification of QTLs for Leaf Traits"

_forests, doi:10.3390/f14081684_

Round 1

Reviewer 1 Report

I have outlined my specific concerns below:

Abstract:

Abstract should be summarized only representing key information and results.

Methods and Results:

The quality control procedures for the SNP-calling process are not clearly explained. It would be helpful to provide information on the thresholds used, such as minor allele frequency (MAF), criteria for missing individuals, and thresholds for missing SNP markers.

SNP markers with missing more than 15% have been excluded? This threshold is not strict enough. Please justify the chosen threshold or consider a more rigorous criterion.

Please provide more details on the phenotyping section. Specifically, describe how the plant samples were established, the age of the plants at the time of sampling, and the growth conditions under which the plants were cultivated.

Why the authors applied the composite interval mapping (CIM) model instead other methods? Justify the suitability of CIM for this study.

Consider incorporating multivariate analysis, such as principal component analysis (PCA) and clustering, on the phenotypic data. This would enhance the analysis and interpretation of the results.

Figure 1 requires improvement as its quality is poor and it lacks informative content. Please ensure that the figures provide clear and relevant information to the readers. 

The visualization of Figures 2 and 3 is inadequate. It is necessary to redraw all the figures in the manuscript to ensure they effectively convey the information to the readers.

Enhance the presentation of the results. The current format does not effectively communicate the findings. Consider restructuring and rephrasing the results section to improve clarity and coherence.

The discussion section is very brief, and the findings of this study are not compared with previous works. Please expand the discussion and provide a comprehensive analysis of the results in the context of relevant literature.

Reviewer 2 Report

The authors constructed a linkage map of chestnut and identified leaf QTLs which can have implications in selection for fruit quality and stress tolerance in this species.

The results are well presented and sufficiently described. The figure quality must be improved to be accepted in this journal.  

Reviewer 3 Report

Overall, the paper addresses an important matter that is mapping QTLs of important traits of chestnut. However, I have to point some major concerns that reflect my decision: 

- GBS has been performed after quantifying DNA samples with Nanodrop only? Did the authors consider to use Qubit fluorometric quantification? That should be the case for such type of sequencing/analyses; 

- Figures have poor quality and it is not possible to read and identify loci. The authors should consider a high quality TIFF figure and with 800-900 dpi resolution or higher, perhaps. Alternatively, a summarized version of each figure, highlight the IDs of the main markers could be presented instead. 

- Several grammar misuses were identified through the entire reading. Please improve English thoroughly. 

- These days, normally three consecutive years and/or environments are advised for reproducible QTL analyses, and the authors conducted two. That is not a decisive point in my final recommendation, but the authors should consider it for future endeavors. 

I believe that the authors need to address the written expression carefully and explain more details on the GBS analyses.

Frequent errors in grammar should be properly addressed by an expert service in English. 

Round 2

Reviewer 3 Report

I think the article has improved significantly, so it can be approved.